# Assessment of staffing needs for physicians and nurses at Upazila health complexes in Bangladesh using WHO workload indicators of staffing need (WISN) method

Taufique Joarder [1,2] Samiun Nazrin Bente Kamal Tune [1] Md Nuruzzaman,[3] Sabina Alam,[4] Valeria de Oliveira Cruz,[3] Tomas Zapata [5]

[1]BRAC James P Grant School of Public Health, BRAC University, Dhaka, Bangladesh
[2]Bangladesh Office, FHI 360, Dhaka, Bangladesh
[3]World Health Organization Bangladesh Country Office, Dhaka, Bangladesh
[4]Ministry of Health and Family Welfare, Government of Bangladesh, Dhaka, Bangladesh
[5]WHO South-East Asia Regional Office, New Delhi, India

**Correspondence to**
Dr Taufique Joarder;
taufiquejoarder@gmail.com

## ABSTRACT

**Objective** This study aimed to assess the current workload and staffing need of physicians and nurses for delivering optimum healthcare services at the Upazila Health Complexes (UpHCs) in Bangladesh.

**Design** Mixed-methods, combining qualitative (eg, document reviews, key informant interviews, in-depth interviews, observations) and quantitative methods (time-motion survey).

**Setting** Study was conducted in 24 health facilities of Bangladesh. However, UpHCs being the nucleus of primary healthcare in Bangladesh, this manuscript limits itself to reporting the findings from the providers at four UpHCs under this project.

**Participants** 18 physicians and 51 nurses, males and females.

**Primary outcome measures** Workload components were defined based on inputs from five experts, refined by nine service providers. Using WHO Workload Indicator of Staffing Need (WISN) software, standard workload, category allowance factor, individual allowance factor, total required number of staff, WISN difference and WISN ratio were calculated.

**Results** Physicians have very high (WISN ratio 0.43) and nurse high (WISN ratio 0.69) workload pressure. 50% of nurses' time are occupied with support activities, instead of nursing care. There are different workloads among the same staff category in different health facilities. If only the vacant posts are filled, the workload is reduced. In fact, sanctioned number of physicians and nurses is more than actual need.

**Conclusions** It is evident that high workload pressures prevail for physicians and nurses at the UpHCs. This reveals high demand for these health workforces in the respective subdistricts. WISN method can aid the policy-makers in optimising utilisation of existing human resources. Therefore, the government should adopt flexible health workforce planning and recruitment policy to manage the patient load and disease burden. WISN should, thus, be incorporated as a planning tool for health managers. There should be a regular review of health workforce management decisions, and these should be amended based on periodic reviews.

## Strengths and limitations of this study

► Time-motion findings helped the experts suggest a more context sensitive activity standards.
► Using both qualitative and quantitative methods for primary data collection complemented each other for bringing data accuracy.
► Technical inputs from the WHO technical experts in Workload Indicator of Staffing Need (WISN) application in other countries, and officials of the Ministry of Health improved the data quality as they were directly involved in quality checks at the field level.
► One limitation was that some service statistics data, essential for establishing standard workloads, were unavailable.
► Due to lack of scope in WISN methodology, patient engagement was minimal.

## BACKGROUND

Shortage of human resources for heath (HRH) has been one of the major challenges faced by the health system and globally, more than 90 countries are haunted by this crisis. According to International Labour Organization, there are on average 34.5 health workers per 10 000 population and about one-third of the world's population lack access to healthcare because of shortage of health workforce.[1] According to the global strategy on HRH: workforce 2030, the estimated global shortage of skilled health workers will be around 18 million by 2030.[2] This problem has reached a critical stage in three (Bangladesh, Myanmar and Bhutan) South-East Asia Region countries with <23 health workers (doctors, nurses and midwives) per 10 000 population, limiting access to health services.[3]

Improving health workers' performance and productivity is vital for better health service provision in the country. Poor

performance of the health workers has been reported in the literature resulting from too few staff, or staff not providing care according to standards.[4–6] The extent of the shortage is reflected in health worker density rates and workforce vacancy rates, and its impact in health system performance indicators. Factors that contribute to poor performance of health workers include limited employment opportunities and low salaries; poor working conditions, weak support and supervision, and limited opportunities for professional development.[7]

Bangladesh's health workforce scenario is characterised by 'shortage, inappropriate skill mix and inequitable distribution'.[8 9] Equitable access to skilled and motivated health worker in a functional health system is essential for achieving Universal Health Coverage and the Sustainable Development Goals.[10] In 2015, Government of Bangladesh approved the Bangladesh Health Workforce Strategy which affirms government's vision of equitable availability of skilled, motivated and responsive health workforce in adequate numbers across the country.[11] However, there is lack of comprehensive, nationally representative data on HRH workload and optimum staff need in healthcare facilities in Bangladesh. A small scale qualitative study found overwhelming workload as one of the critical components that hinders retention of doctors and nurses at rural healthcare facilities in Bangladesh.[12] Another policy analysis on retention of HRH (physicians and nurses) also found that deficiency of adequate workforce and consequent high workload acted as a deterrent against rural retention.[13]

Workload management is very important for any country or institution to deliver quality services, retain staffs and reduce turnover.[14] Even the seminal document on HRH, 'Global strategy on HRH: Workforce 2030', emphasised on low/middle-income country (LMIC) level workforce strategies, drawing on workload analysis studies.[2] Such studies can provide detailed insight into the current state of workload in a system, coping strategies of the staff for regular extra work pressure, causes behind the excessive workloads, and ways to deal with it. This study aimed to fill-in this knowledge gap with respect to workload and optimum staff need for physicians and nurses at the Upazila or subdistrict level (ie, at Upazila Health Complex (UpHC)). It is expected that this workload analysis will contribute in improving performance, ensuring quality of services and facilitating uninterrupted service delivery through efficient management of staff.

## Workload Indicator of Staffing Need overview

WHO developed the Workload Indicator of Staffing Need (WISN) method in 1998, which was later updated based on learning from implementation in different countries. This method is simple, useful and time-saving, which was borrowed from the industrial sector for use in the health sector by Peter Shipp in 1984. The result is expressed in terms of differences and ratios, the former indicating worker shortage or surplus, and the latter workload pressure experienced by the staff.[15]

**Figure 1** The ways WISN can help in human resource decision making. WISN, Workload Indicator of Staffing Need.

WISN results help in human resource decision making in several ways (figure 1). For example, recruitment and transfer of HRH can be based on geographical comparison of WISN ratios, staffing of health facilities can be informed by WISN-based workload projection.

## METHODS
### Study design
We followed the updated WISN manual,[15] but contextualised it for Bangladeshi setting. The WISN steps have been summarised in figure 2. The research project was developed based on close collaboration among and mutual insights from three types of committees:

### Steering committee
The steering committee (SC) was consisting 13 members, established by the Ministry of Health and Family Welfare (MOHFW) with membership from senior government officials (seven): WHO official (one); professional organisation of the physicians, Bangladesh Medical Association (one) and relevant academia (four) such as BRAC University (two persons), Bangladesh University of Health Sciences and Centre for Medical Education. All seven senior government officials were directly involved in decision making regarding daily management of the health workforce in their respective departments. WHO officer was there to respond to WISN-related technical issues and application. All four academicians were part of the committee because they had expertise in their respective areas (ie, education, policy-making, curriculum development and performance assessment) of the health workforce. They were also well known in the community of scientific writing and academic teaching. The role of the SC was to guide and endorse the overall study based on the WISN strategy and its implementation.

**Figure 2** Methods applied in each WISN step. WISN, Workload Indicator of Staffing Need.

### Technical task force

Technical task force (TT) was responsible for guiding the implementation of the WISN process. Researchers from implementing research institution (the school of public health of a Bangladeshi university, BRAC University); and experts from WHO Bangladesh Country Office; an international non-governmental organisation, Save the Children and another university, Bangladesh University of Health Sciences served in the task force.

### Expert working groups

There were multiple expert working groups (EWGs), one for each of the following professional groups: general physicians (medical officer (MO), emergency medical officer (EMO), residential medical officer (RMO)) and nursing staff (senior staff nurse, nursing supervisor). The respective EWG defined the workload components and set activity standard for the specific staff category.

The qualitative part of this research involved document reviews, key informant interviews (KIIs) with policy-level persons related to HRH issues in Bangladesh (mostly

from among SC and EWG members), in-depth interviews (IDIs) with individual service providers (eg, physicians, nurses, etc working in UpHCs under this study) and observations. The quantitative component involved time–motion survey, which is a work measurement technique for recording the times and rates of working for the elements of a specific job though observing a subject continuously or in a certain period of time.[16] Time and motion data served as a guide to determine the activity standard for WISN analyses.

### Study duration, setting and population

The research continued from July to November 2017 and was carried out in two preselected districts of Bangladesh (figure 3).

► Jhenaidah, located in Southwestern part of Bangladesh, under Khulna division.
► Moulvibazar, located in Northeastern part of Bangladesh, under Sylhet division.

The selection of the districts was made by the Human Resources Unit, MOHFW in collaboration with the

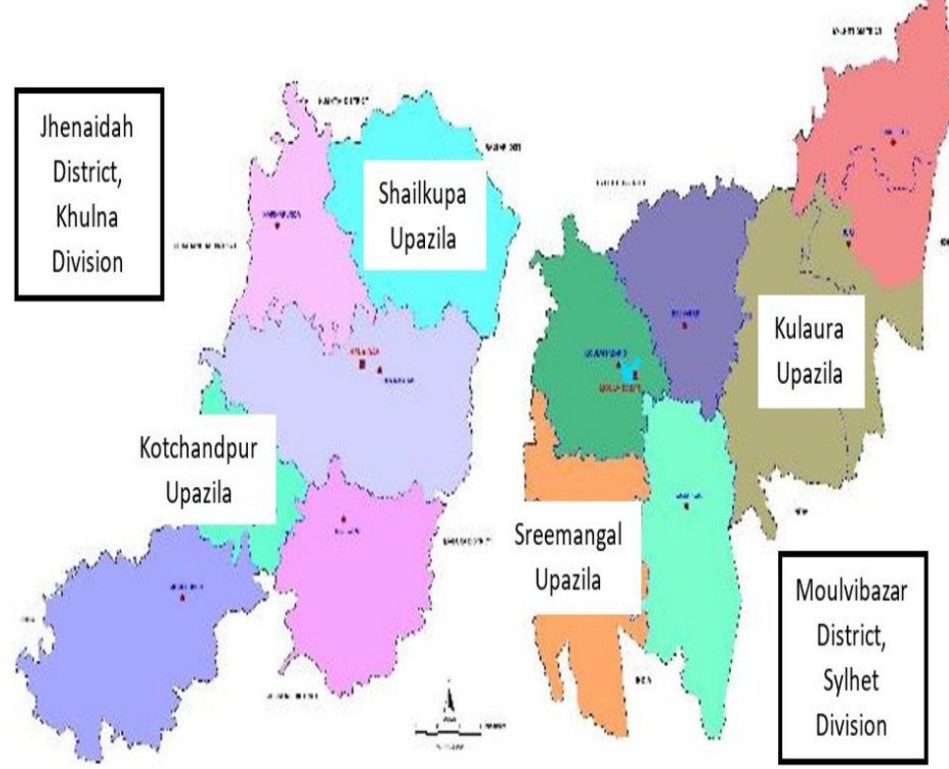

**Figure 3** Map of Jhenaidah and Moulvibazar districts with study Upazilas identified.

development partners (Save the Children and WHO Bangladesh Country Office) in a workshop nationally organised in November 2016. Later the selection of the UpHCs was made in consultation with the SC, taking into consideration some performance indicators such as number of beds, number of total deliveries, number of live births, bed occupancy rates as well as patient load including the number of outpatient visits. From each district, two highest performing UpHCs were included in the study.[17] Thus, Shailkupa and Kotchandpur UpHCs were selected from Jhenaidah district and Kulaura and Sreemangal from the Moulvibazar. Because future applications need to be based on optimum quality standards, the highest performing UpHCs were selected to serve as a model for other health facilities. From each of these four UpHCs, physicians (MOs and RMOs) and nurses (senior staff nurse and nursing supervisor) were included for workload analysis. A total of 24 health facilities from the two districts were studied. This included two district hospitals, four UpHCs, two Maternal and Child Welfare Centres, eight Union Sub Centres, eight Union Health and Family Welfare Centres and eight Community Clinics. This manuscript reported findings from all the four UpHCs under this study, as UpHCs are the nucleus of the primary healthcare delivery in Bangladesh, serving the rural population.

## Sampling strategy
### Qualitative part
Documents for review were selected based on the suggestions from the experts (members of SC, TT and EWG), supplemented by reference tracking of government reports and published literature on HRH of Bangladesh. Key Informants were selected on the principles of purposive sampling,[18] supplemented by snowball sampling (ie, based on the reference or suggestion from the key informants). IDI respondents were selected through purposive sampling, based on the respondent's seniority and designation (eg, RMO, Nursing Supervisors, etc). These respondents were practising individuals and had more than 10 years of experience and played a supervisory role in their respective health facilities.

### Quantitative part
For time-motion study, time sampling was done for each consenting staff available during the data collection period. Field data collectors (FDCs) observed each staff twice for 45 min duration, once during the first half of their service duration and again during the second half. This was done to minimise the bias in the time data due to the patient load (assuming higher patient load in the first half and lower in the second).

### Tool development, pretest, training of data collectors, agreement test
For qualitative data collection, semistructured guidelines, including that for document reviews, KIIs, IDIs and observation, were developed. For the time-motion study, structured observation tool was designed. The structured observation tools contained three sections:

1. Background information of observation setting and the person under observation.
2. Time–motion data sheet (containing three columns: type of activity, time spent in minutes and remarks).
3. Example of the activities (health service activities, support activities and additional activities).

Health service activities, according to WISN manual, are performed by all members of the staff category and regular service statistics are available for them, for example, obstetrical service, emergency service, outpatient service. Support activities are also performed by all members of the staff category, but regular service statistics are not available for them, for example, record keeping and reporting, attending meetings, instrument sterilisation. Additional activities are performed by only certain members of the staff category (eg, the supervisor or a senior member), and regular service statistics are not available for them, for example, duty roster preparation, preparing staff evaluation reports, supervision of cleanliness.[15]

The examples of health service activities were primarily drawn from the list of the activities mentioned in the Essential Services Package (ESP)[19] for respective health facility type. Since service statistics were not available according the ESP activity list, it was adjusted for the local context with inputs from the respondents (through KII, followed by IDI), in alignment with the availability of service statistics. An 'hourglass' approach was adopted for defining the workload components based on the ESP (figure 4). Tools were pretested in a UpHC near Dhaka, before applying for actual data collection. Qualitative tools were also pretested through mock IDIs and KIIs. The pretesting exercise was followed by the training of the field supervisors (FSs) and FDCs.

**Table 1** Number of interviewees representing EWG of staff categories

| EWG representing staff category | No |
| --- | --- |
| Physicians: MO, EMO, RMO | 2 |
| Nurses: nursing supervisor, senior staff nurse | 7 |
| Total | 9 |

EMO, Emergency Medical Officer; MO, Medical Officer; RMO, Residential Medical Officer.

### Data collection and quality control

At first, FSs were sent to respective districts to orient the personnel on the project, seek support and assess the availability of the service statistics. The FSs spent 1 week in each district and conducted qualitative observation of the service provision at the UpHCs, to gain a firsthand understanding of the context.

In the third step, the FDCs, under the supervision of FSs, conducted time-motion study, using a mobile device (SurveyCTO software). During the time, FDCs were collecting time-motion data, the FS, in addition to supervising the FDCs, conducted additional IDIs and collected data on available working time; time required for health service, support and additional activities; and service statistics.

The next step was data validation and set activity standards. Primary data validation was done through phone calls made to the services providers, and health facility statisticians. Second, these were shared with the SC and TT members. Finally, interviews were conducted with the EWG members to finalise the activity standards (table 1).

We presented the time-motion findings to the EWG members and requested them to be as realistic as possible in suggesting activity standards. We also requested them to account for the variability of patient load due to factors such as seasonality, timing of day, facility catchment population. Finally, through multiple meetings, debates and deliberations, the activity standard was finalised, taking into account the information from the IDIs and KIIs, and inputs from the EWG members (table 2).

In order to ensure the quality of data, the principal expert (lead author of this article), coexperts (two coexperts—one was leading data collection and the other was leading data quality check and reporting), WHO team consisting of national and international technical experts, and officials from the Human Resource Branch of MoHFW conducted field visits to each study District and the health facilities therein. During the time–motion data collection period, the coexperts monitored the data and their geographical location in real time. They also regularly checked the consistency of the data. Our field based data collection team saved contact information of all the respondents; so, in case of any confusion or need for clarification, the coexperts called the respondents over phone.

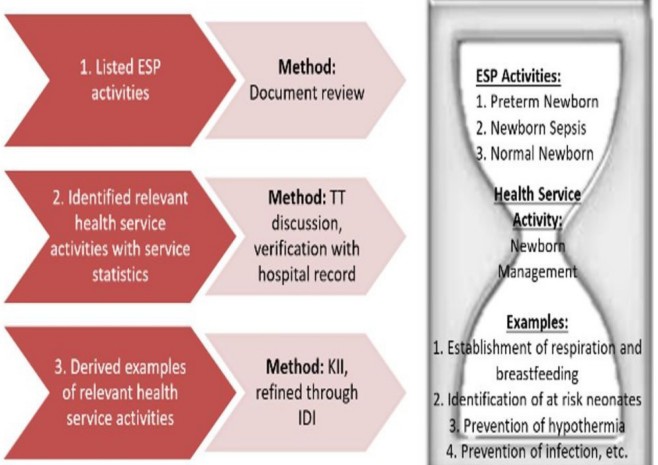

**Figure 4** Approach of integration of ESP components in defining workload components of health service activity. ESP, Essential Services Package; TT, Technical Task Force; IDI, in-depth interview; KII, key informant interview.

**Table 2** Service standard for physicians and nurses in UpHCs, Bangladesh, 2017

| Activities | Service standard | | Unit |
|---|---|---|---|
| | Physician | Nurse | |
| Obstetrical service (caesarean section)* | 90 | 90 | Min/patient |
| Obstetrical service (normal delivery) | 60 | 120 | Min/patient |
| Newborn management | 15 | 15 | Min/inpatient day |
| Emergency service | 15 | Not applicable | Min/patient |
| IMCI/nutritional service | 15 | 15 | Min/patient |
| OPD service (including NCD management) | 10 | Not applicable | Min/patient |
| First ANC | 20 | 20 | Min/patient |
| Follow-up ANC | 10 | 10 | Min/patient |
| PNC | 15 | 15 | Min/patient |
| Indoor services (round, including minor bedside procedures)† | 25.65 | 34 | Min/inpatient day |
| Bedside patient care | Not applicable | 17 | Min/inpatient day |
| Patient admission and discharge | Not applicable | 20 | Min/inpatient day |
| Death certification and associated arrangements | 20 | 30 | Min/patient |

*For nurses: assist obstetrical service (caesarean section).
†For nurses: indoor services (round with physician).
ANC, antenatal care; IMCI, Integrated Management of Childhood Illness; NCD, non-communicable diseases; OPD, out patient department; PNC, postnatal care; UpHCs, Upazila Health Complexes.

## Data management and analysis

The first analytical step was to estimate available working time of the staffs. This is the time a health worker has available in 1 year to do his or her work, taking into account authorised and unauthorised absences.[15] For all categories of staff, a uniform number of weeks per year (52 weeks), working days in 1 week (6 days), possible working days in 1 year (52×6=312 days) were estimated. Next, absent days, such as public holidays (20 days), earned leave (average for each staff category, based on Health Management Information System data) and casual leave (20 days) were deducted to obtain the annual working time in days. Multiplying this with daily working hours (6 hours per day), we obtained annual working time in hours.

Workload components were defined through the inputs from the key informants; activity standards were also set through the interviews with the EWG members. An activity standard is the time necessary for a well-trained, skilled and motivated worker to perform an activity to professional standards in the local circumstances.[15] Both service standards (for health service activities), category allowance standards (for support activities) and individual allowance standards (for additional activities) were determined in the same way.

The next analytical step was to establish standard workload, which was done by dividing the annual working time by unit time of health service activities. A standard workload is the amount of work within a health services workload component that one health service provider can do in a year hypothetically.[15] Then category allowance factor and individual allowance factors were calculated using the following formula, respectively:

Category allowance factor=1/{1−(Total category allowance standard/100)}

Individual allowance factor=Total individual allowance standard/Available working time in hours

Next, exact number of required staff was calculated by the following formula:

Total required number of staff=(Staff needed for health service activity * Category allowance factor)+Individual allowance factor

The fractional results were rounded up or down, following the guideline provided in the WISN manual[15]:

► 1.0–1.1 is rounded down to 1 and >1.1–1.9 is rounded up to 2.
► 2.0–2.2 is rounded down to 2 and >2.2–2.9 is rounded up to 3.
► 3.0–3.3 is rounded down to 3 and >3.3–3.9 is rounded up to 4.
► 4.0–4.4 is rounded down to 4 and >4.4–4.9 is rounded up to 5.
► 5.0–5.5 is rounded down to 5 and >5.5–5.9 is rounded up to 6.

Finally, based on the existing number of staff in the respective health facilities, we calculated both the difference (current number of staff—required number of staff by WISN), and the ratio (current number of staff/required number of staff by WISN). The WISN difference indicates whether the health facilities are relatively understaffed (ie, when the WISN difference is negative), overstaffed (ie, when the WISN difference is positive) or balanced (ie, when the WISN difference is 0). The WISN ratio indicates whether the staffs are experiencing high workload (ie, when the WISN ratio is lower than 1), low workload (ie, when the WISN ratio is higher than 1) or normal workload (ie, when the WISN ratio is equal to 1). For this calculation, we used the de facto number of current staff, that is, the number of staff that we actually found working in the health facilities during our data collection period; not the number shown in the office records or statistics.

We obtained a support letter from the Ministry of Health and Family Welfare and also obtained written informed consent form from each person we engaged in data collection. Identity of respondents was kept confidential.

**Table 3** Analysis of WISN results at aggregate level (average required number and WISN ratio across same types of health facilities)

| Staff category | Required staff to cope with the demand | Average no of existing staff | Deficit of staff | Average WISN ratio | Workload pressure |
|---|---|---|---|---|---|
| Physician | 10.59 | 4.50 | 6.09 | 0.43 | Very high |
| Nurse | 18.86 | 12.75 | 6.11 | 0.69 | High |

WISN, Workload Indicator of Staffing Need.

## RESULTS
### General WISN findings across levels

For descriptive purposes, we have categorised the workload pressure as Extremely High (WISN ratio between 0.10 and 0.29), very high (WISN ratio between 0.30 and 0.49), high (WISN ratio between 0.50 and 0.69), moderately high (WISN ratio between 0.70 and 0.89), normal (WISN ratio between 0.90 and 1.19) and low (WISN ratio greater than or equal to 1.20). Based on this categorisation, at an aggregate level (ie, considering the average required number and WISN ratio across the same types of health facilities), physicians are found to have a very high (WISN ratio 0.43) and nurses high (WISN ratio 0.69) workload pressure. To cope with the workload, on an average 11 physicians (on average 4.50 were available during data collection) and 19 nurses (On average 12.75 were available during data collection) are needed in each UpHC (table 3). This means, there was an average deficit of six members in each staff categories.

Tabulating the total percentage of time spent on all support activities (ie, category allowance standards) by different staff categories, we found that, 50% of nurses' time are occupied with support activities (table 4).

### WISN results disaggregated by UpHCs

The required number of staff ranges from 8 to 12 among physicians, and 16 to 23 among nurses. Highest shortage is observed in nurses of Sreemangal UpHC (−8.46), followed by physicians of Kulaura UpHC (−8.28). Workload pressure is the highest among physicians of Kotchandpur UpHC (WISN ratio 0.28) and lowest among nurses of Shailkupa (WISN Ratio 0.87) (table 5).

### Change of workload if vacancies are filled

If the vacant posts are filled, understandably, the workload is reduced. In most of the cases, sanctioned number of physicians and nurses is more than what is actually needed to tackle the workload. However, only filling up the vacant posts are not enough in case of some of the

**Table 4** Comparison of support activities across staff categories

| Staff category | Total % of support activities |
|---|---|
| Physician | 24 |
| Nurse | 50 |

staff categories, such as the nurses at Kotchandpur and physicians at Sreemangal (table 6).

## DISCUSSION AND RECOMMENDATIONS
### Discussion

Findings from this WISN study clearly indicates that the public sector healthcare providers in Bangladesh are suffering from a very high workload pressure. Nurses are predominantly occupied with support activities rather than actual nursing care. There is unequal workload across UpHCs, indicating potential for workforce redistribution. The unequal workload mainly stems from differing patient load, due to geographical location, number of catchment population and epidemiological characteristics, at different UpHCs. Inappropriate number of sanctioned posts indicate the necessity of WISN-based workforce planning.

High workload pressure may arise from absolute or relative shortage of health workforce. Absolute shortage appears when there is inadequate production of a particular staff category while relative shortage appears when health workforce is not distributed evenly between the urban and rural areas throughout the country for various reasons. For example, absolute shortage in HRH production is revealed by the fact that there are only 4.90 registered physicians and 2.90 registered nurses per 10 000 populations,[17] rendering the country to be one of the 57 critical workforce shortage countries in the world.[6] On top of this absolute shortage, Bangladesh also suffers from relative shortage, as evidenced from the fact that physician to population ratio in urban areas is 1:1500, but in rural areas it is 1:15 000.[20] Workload pressure has some serious consequences as well, namely, fatigue and burnout of service providers, lack of motivation and compromised quality of care.[21] High workload is, however, not unique to Bangladesh. WISN studies in LMICs like Namibia,[22] Uganda,[23] Kenya,[24] Burkina Faso[25] and Iran[26] also identified high workload pressure among their HRH.

It is expected that the nurses would spend most of their service times beside the patients, providing nursing care. Unfortunately, this is not the case in Bangladesh as well as in some other comparable settings. A qualitative study in Bangladesh showed that nurses' maximum time is spent on administrative and paperwork tasks.[27] Excessive support activities of nurses are reported in studies

**Table 5** Analysis of WISN results of Upazila-level health staff

| Health facility | Current no of staff | Required no, based on WISN | Shortage or excess | WISN ratio | Workload pressure |
|---|---|---|---|---|---|
| Staff category: physician | | | | | |
| Shailkupa UpHC | 4 | 8.14 | −4.14 | 0.49 | Very high |
| Kotchandpur UpHC | 3 | 10.71 | −7.71 | 0.28 | Extremely high |
| Kulaura UpHC | 4 | 12.28 | −8.28 | 0.33 | Very high |
| Sreemangal UpHC | 7 | 11.23 | −4.23 | 0.62 | High |
| Staff category: nurse | | | | | |
| Shailkupa UpHC | 14 | 16.08 | −2.08 | 0.87 | Moderately high |
| Kotchandpur UpHC | 15 | 22.80 | −7.8 | 0.66 | High |
| Kulaura UpHC | 10 | 16.08 | −6.08 | 0.62 | High |
| Sreemangal UpHC | 12 | 20.46 | −8.46 | 0.59 | High |

UpHC, Upazila Health Complex; WISN, Workload Indicator of Staffing Need.

conducted in Iran[26] and Uganda[23] as well. A recent WISN study conducted in Iran showed that nurses are overburdened; and support activities account for 31% of their workload.[26] Nurses' excessive engagement in paperwork or other support activities may result from deficient human resource planning and management.

Despite the fact that most of the staff are already overworked, staffs in some health facilities may be more so, compared with a neighbouring one. Presence of different number of staffs causes fluctuation in the amount of workload at different health facilities. In places where workload of a staff category is 'extremely high', some supports from nearby health facilities with lower workload should be sought. Or, in places where workload of a staff category is 'normal' or 'low', some support may be transferred to health facilities with higher workload. For example, in Sreemangal UpHC, there are seven physicians, with a high workload pressure. However, workload pressure in nearby Kulaura UpHC is very high, with only four physicians (table 5). At least one physician from Sreemangal can be reallocated to Kulaura to tackle the high workload. Similar action may be taken regarding the nurses by transferring some from Shailkupa (moderately high workload) to Kotchandpur (high workload). This is just an example how WISN can help in decision making regarding allocation of human resources. Similar situation was identified in Namibia, where researchers suggested redistribution of health workers from one area to the other.[22]

We found that many posts remained vacant in different health facilities. Some staffs were not present at their service locations for various reasons, such as training, deputation to another health facility. Even if the existing posts are filled up, a large portion of the workload would be curbed. For example, according to the Standard Setup document of the Ministry of Public Administration, 18 physician posts (10 junior consultants, 1 RMO, 7 assistant surgeons) have been proposed for a 50 bed hospital.[28] We have found 4.5 physicians on an average in each UpHC. The average required number is 11 (table 3). Our proposition is that, even if it is not possible to reach the ideal workforce setup for a health facility, filling up at least the vacant positions and ensuring regular presence of all staffs would reduce the workload. Supportive supervision

**Table 6** Change of workload if vacancies in physician and nursing posts are filled

| Health facility | Staff category | Current no of staff | Required no, based on WISN | WISN ratio | Sanctioned no of staff | WISN ratio as per sanctioned no of staff |
|---|---|---|---|---|---|---|
| Shailkupa UpHC | Physician | 4 | 8.14 | 0.49 | 10 | 1.25 |
| | Nurse | 14 | 16.08 | 0.87 | 21 | 1.31 |
| Kotchandpur UpHC | Physician | 3 | 10.71 | 0.28 | 20 | 1.82 |
| | Nurse | 15 | 22.80 | 0.66 | 20 | 0.87 |
| Kulaura UpHC | Physician | 4 | 12.28 | 0.33 | 20 | 1.67 |
| | Nurse | 10 | 16.08 | 0.62 | 26 | 1.62 |
| Sreemangal UpHC | Physician | 7 | 11.23 | 0.62 | 10 | 0.91 |
| | Nurse | 12 | 20.46 | 0.59 | 22 | 1.10 |

UpHC, Upazila Health Complex; WISN, Workload Indicator of Staffing Need.

and monitoring of the staff is essential to ensure the presence of posted staff. Researchers in Namibia came up with the similar finding and proposed a similar solution for the problem.[22] WISN was used over standard staffing schedule in HIV Clinics in Kenya as well to resolve a similar crisis.[24]

## Recommendations

Based on the findings and its in-depth analysis, we propose few short-term and long-term recommendations. The short-term recommendations require administrative or management decisions, relatively easier to implement and tackle the immediate crisis. On the other hand, the long-term recommendations demand radical policy amendments following careful examination.

Short-term recommendations include: reallocation of staff from low workload areas to high workload areas, fill up existing vacant positions and strengthen supervision and monitoring. Nurses are the most needed staff, the most overloaded and are short in supply. On top of all these, they are burdened with support activities. If some of their support and additional activities can be shifted to other staff, nurses can devote their time better in nursing care.

The study yields some long-term recommendations as well, for the policy-makers. For example, in order to increase the availability of workforce, especially nurses, and decrease their workload, their number needs to increase. Hence, long-term policy response is needed to increase the intake of nursing students, train them with quality education and deploy them in larger numbers in a secure and gender-friendly work environment. In the same vein, incentives should be given to increase the number of nurses in both public and private sector educational institutions. Regulations should be developed and implemented so that medical colleges can be established only when a nursing school is established alongside. Otherwise, the skill-mix imbalance between physicians and nurses would jeopardise the quality of care. Quality and quantity of physicians should also increase.

Second, since the nurses are found to be predominantly engaged in support activities at the expense of actual patient care, a separate staff category for administrative/support activities is greatly warranted. This will free up the valuable yet scarce clinical time of the service providers.

Third, instead of the existing approach of deploying a fixed number of workforce at all health facilities, a flexible recruitment and HRH planning is needed, based on patient load and disease burden. This can be supported by determination of optimal requirements of HRH under given limited resources or constraints in those health facilities[29–32] by using the WHO methodology on Workload Indicators of Staffing Need. It is important to recognise that decisions in health sector are very much contingent on the local context, especially the patient load, demographic drivers (eg, age structure of the population, gender ratio, etc) and epidemiological profile.

Therefore, the government should adopt flexible health workforce planning and recruitment policy in place to keep up with the local patient load and disease burden. The culture of bottom-up decision making should be adopted eventually.

## Strengths and limitations

This study had a number of strengths. First, we conducted time-motion study, which helped the research team gain a better understanding of the service context of the staffs. Second, when the key informants or experts suggested an unrealistic activity standard, we presented them the time-motion findings and helped them suggest a more context sensitive standards. Third, the research team used both qualitative and quantitative methods for primary data collection, which complement each other for bringing data accuracy. Fourth, WHO technical officers, who had expertise in WISN application in other countries, and the Ministry of Health and Family Welfare officials were directly involved in the field level data quality checks.

However, despite careful planning and painstaking implementation of the research, we faced some challenges during different stages of the WISN process. First, some service statistics data, which were essential for establishing standard workloads, were not readily available due to poor record keeping systems at some health facilities. Second, the research did not take into account the patients' opinion or stakeholders' stance. Notwithstanding the fact that these perspectives are gaining momentum in health workforce decision making, we could not take advantage of them in the interest of adhering to the highly structured nature of the WISN methodology. Third, the official number of existing staffs often did not match with the number of staffs we observed providing services.

## CONCLUSIONS

Human resource management is a big challenge, especially in a resource-poor setting like Bangladesh. With a vision of becoming a middle-income country by 2021, Bangladesh needs to strive for optimising its existing resources, including human resources. This type of study can aid the decision making in this direction, using the WISN as a planning tool for the managers. Implementation research is needed regarding how this workload-based staffing decisions can be integrated into the health systems in the most effective way. We expect that these types of studies would pave the way for evidence-based HRH decision making in the context of health system of Bangladesh.

## Consent for publication

During the data collection process, while the ethical consents were obtained from the respondents, they were informed that their data might be used for publication in future. They were also informed that their identity will remain anonymous. Institutional consents for publication were obtained as well.

**Acknowledgements** The authors express their gratitude to BRAC James P Grant School of Public Health, BRAC University; the WHO Country Office, Bangladesh for their support and contributions in this research effort, as well as the MoHFW for supporting the research team throughout the study period. We are also thankful to the study participants who consented to participate in the study. We are indebted to Professor Syed Masud Ahmed, who served as the Advisor to the study project, and provided valuable technical inputs at different stages of the project. Finally, we would like to extend our sincere gratitude to the Field Supervisors and Data Collectors for their contributions throughout the qualitative and quantitative data collection.

**Contributors** TJ conceived and designed the study. TJ and SNBKT carried out the data analyses and drafted the manuscript. MN, SA and VdoC provided substantial technical inputs in the inception phase and throughout the research process. MN, SA, VdOC and TZ thoroughly reviewed the manuscript and contributed substantially with the necessary revision. TJ and SNBKT again reviewed the manuscript and prepared for the final submission. All authors approved the final manuscript.

**Funding** World Health Organization Bangladesh Country Office

**Map disclaimer** The depiction of boundaries on this map does not imply the expression of any opinion whatsoever on the part of BMJ (or any member of its group) concerning the legal status of any country, territory, jurisdiction or area or of its authorities. This map is provided without any warranty of any kind, either express or implied.

**Competing interests** None declared.

**Patient consent for publication** Not required.

**Ethics approval** Ethical approval for this study was obtained from the Institutional Review Board (IRB) of BRAC James P Grant School of Public Health, BRAC University. We strictly adhered to all ethical principles.

**Provenance and peer review** Not commissioned; externally peer reviewed.

**Data availability statement** Data are available on reasonable request. All research data have been submitted to WHO Country Office for Bangladesh, as per the agreement with the research organisation, BRAC James P Grant School of Public Health, BRAC University. This manuscript only used the data pertaining to the physicians and nurses working at the two study Upazila (subdistrict) Health Complexes. Unpublished data include: consultants, general physicians and nurses at the district hospital level; physicians, and Family Welfare Visitors at Maternal and Child Welfare Center level; subassistant community medical officers at the Upazila Health Complex level; subassistant community medical officers at Union SubCenters level, subassistant community medical officers, and Family Welfare Visitors at Union Health and Family Welfare Centers level and Community Health Care Providers, and Family Welfare Assistants at the Community Clinic/ Outreach level. Data may be obtained from the World Health Organization Bangladesh Country Office on reasonable request (Focal Point: NPO-HRH- nuruzzamanm@who.int).

**ORCID iDs**
Taufique Joarder http://orcid.org/0000-0002-3299-2628
Samiun Nazrin Bente Kamal Tune http://orcid.org/0000-0003-0308-2817
Tomas Zapata http://orcid.org/0000-0002-7807-3520

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
