## [Reviewer comments · BMJ Open]

ARTICLE DETAILS

TITLE (PROVISIONAL)	An assessment of staffing needs for physicians and nurses at Upazila Health Complexes in Bangladesh using WHO workload indicators of staffing need (WISN) method
AUTHORS	Joarder, Taufique; Tune, Samiun; Nuruzzaman, Md; Alam, Sabina; Cruz, Valeria; Zapata, Tomas

VERSION 1 - REVIEW

REVIEWER	Claire Su-Yeon Park Center for Econometric Optimization in the Nursing Workforce, Seoul, Korea
REVIEW RETURNED	30-Oct-2019

GENERAL COMMENTS	To the authors, Thank you for your hard work. I did my best to give you constructive comments. I hope that my input is helpful for you to make your manuscript perfect. Below I outline an additional place the manuscript needs to be improved. Recommendations Page 24, Line 454: "Thirdly, instead of the existing approach of deploying a fixed number of workforce at all health facilities, a flexible recruitment and HRH planning is needed, based on patient load and diseases burden. This can be supported by determination of absolute requirement of HRH in those health facilities." I agree with your statement; however, I would like to suggest you a little more revision on the sentence I underlined, like this "This can be support by determination of optimal requirements of HRM under given limited resources or constrains in those health facilities (please cite relevant references)." The reason why I ask you to revise it like that is that the "absolute requirement" in the workforce planning cannot guarantee a patient-perceived reasonable quality of care, particularly, in a state of emergency. My First Review Comments Dear Editor, Thank you so much for giving me the opportunity to review the manuscript entitled "An assessment of staffing needs for physicians and nurses at Upazila Health Complexes in Bangladesh using WHO workload indicators of staffing need (WISN) method." This manuscript perfectly falls within my specialization and research domain. I very much enjoyed
---

reviewing this manuscript. This exciting experience inspired me to contemplate the scholarly identity, vision and promise of my own program of research. Even though I participated in this peer review process as a reviewer, I have benefited from such a well-delineated advanced scientific statement. I am truly honored beyond all measure to be a part of the panel of reviewers for your highly-reputed journal, BMJ Open.

This manuscript is significant in terms of (1) conducting a context-sensitive, time-motion study to provide a better understanding of the service context of the staffs in the real-world situations, based on the real-time data collection and analyses; (2) applying a bottom-up approach to determine activity standards (i.e. the authors estimated actual workloads); (3) considering an even distribution as well as an absolute number of physicians and nurses; and (4) revealing the novel finding that “nurses’ maximum time is spent on administrative and paperwork tasks” (pp. 21, line 361-362), which has been less explored/addressed in the previous literature of the nursing workforce mainstream research. Based on the rationales, my recommendation is that BMJ Open would offer authors a revision opportunity.

To the authors,
Thank you for your hard work. I did my best to give you constructive comments. I hope that my input is helpful for you to improve the manuscript. Below I outline additional places the manuscript needs to be improved.

Introduction: Excellent description

Page 2, Line 29: Why four UpHCs, not 24 health facilities of Bangladesh? Please provide a core reason why authors reported the findings from the providers at four UpHCs only.

Page 2, Line 42: Why “there should be a gradual policy shift”? It seems illogical. Please consider to delete the sentence. Otherwise, paraphrase it, please.

Page 2, Line 43: What is “HRH”? Provide the full term at it first use, please.

Page 2, Line 44: Please insert “thus” in the following sentence: i.e. “WISN should thus be incorporated as a planning tool for health managers.”

Page 6, Line 130: What is “NGO”? Provide the full term at it first use, please.

Page 6, Line 124-135: Please provide a specific number of experts per category; e.g., “SC was selected from ** out of *** senior government officials, ** out of *** WHO personnel, and relevant academia (be specific. Which academia? How many experts were recruited from the relevant academia? Why?).” Also, please provide a rationale on your decision (i.e. justification) in terms of a number of experts and the experts’ specialization.

Page 7, Line 153: Is there any special reason why you chose the highest performing UpHCs?

	Page 8, Line 166: Please suggest a possible specific rationale—e.g., working period/years—rather than your assumption, “assuming better knowledge of the activity standards,” to ensure a scientific integrity of the paper. Page 8, Line 184: Please insert “or vice versa” in the end of the sentence. Page 9, Line 186-190: Please provide specific examples per each category, i.e., health service activities, support activities, and additional activities. Page 12, Line 236-243: Who is the principle experts and co-experts, respectively? Please specify them. Too many categorized participants involved in your study makes readers confused. Page 21, Line 381-382: “Some staffs were not present at their service locations for various reasons.” Please provide the specific reasons or examples for BMJ Open readers’ easier and better understanding of the context of your study. Discussion: The authors virtually determined the differences between current and required staffing levels on the basis of actual activity standards (current) and healthcare professionals-perceived activity standards (required) only. That is, this study did not encompass patients’ position (i.e. patient acuity or patient dependency) and stakeholders’ stance (ensuring efficiency [cost] while guaranteeing effectiveness [quality of care]) in the healthcare workforce decision-making. The current research trend in the healthcare workforce resides in presenting informed shared decision-making rationales where parties of interests can be all satisfied in their decision-making. Patient engagement, patient reported outcome measures, precision medicine/nursing, creating shared value, and/or value chain are becoming more and more important for those reasons. All but nursing workforce studies rarely utilized well-verified specific workload indicators in the real-world situations such as the Workload Indicator of Staffing Need (WISN) method; thus, this study is significant and worthwhile. However, BMJ Open readers could benefit more from such a well-developed discussion while BMJ Open editors may find the manuscript of enough import to merit publication in the journal. On this, I would like to suggest the authors additional literature reviews on the issues I mentioned above and integrate them into the authors’ discussion.
--	--

REVIEWER	Serebe A. Gebrie Purdue university Global, Indianapolis, USA.
REVIEW RETURNED	12-Nov-2019

GENERAL COMMENTS	Previous comments have been well addressed. It can be accepted for publication after further explanation added to the ethical process used. In general, it is a high-quality study which can be explained by Mixed study, include experts, and used sequential multiple stages. This study contextually toiled WHO tool(WISN). Previous
---

	comments have been addressed and the paper is well improved now. Abstract – it is good Background Page 9 L162- 168, Avoid the bold font. Ethical considerations: L 324 , “Appropriate consent process...” I think it is better to put the actual consent process that you have used. For example, Approval and support letter from ministry of health and wellness may be needed. Then how you did proceed to the health facilities, and KI should be explained. Discussion: In this study, it is mention that nurses overburdened by supportive activities such as paper works, documentation, scheduling. However, such activities are parts of providers duty and they have role on quality and safe patient care provision. Do you think such tasks can be shifted to non-clinical staff ? Who can take over these supportive activities? Please add your opinion to the discussion or recommendation section. Competing interest WHO is the funder of this project and I see author/s from WHO, Bangladesh Office? Is it possible to declare that the authors have no competing interest?
--	--

VERSION 1 – AUTHOR RESPONSE

Reviewer 1:

RECOMMENDATIONS:

Page 24, Line 454: “Thirdly, instead of the existing approach of deploying a fixed number of workforce at all health facilities, a flexible recruitment and HRH planning is needed, based on patient load and diseases burden. This can be supported by determination of absolute requirement of HRH in those health facilities.” I agree with your statement; however, I would like to suggest you a little more revision on the sentence I underlined, like this “This can be support by determination of optimal requirements of HRM under given limited resources or constrains in those health facilities (please cite relevant references).” The reason why I ask you to revise it like that is that the “absolute requirement” in the workforce planning cannot guarantee a patient perceived reasonable quality of care, particularly, in a state of emergency.

Response: Thanks for the feedback. We have edited the sentence accordingly.

Reviewer 2:

BACKGROUND:

Page 9 L162- 168, Avoid the bold font.

Response: We carefully reviewed the manuscript and could not find the bold texts in the submitted manuscript. This often happens when the Word document is converted into a PDF file.

ETHICAL CONSIDERATIONS:

L 324 , “Appropriate consent process...” I think it is better to put the actual consent process that you have used. For example, Approval and support letter from ministry of health and wellness may be needed. Then how you did proceed to the health facilities, and KI should be explained.

Response: We have updated the section as suggested.

DISCUSSION:

In this study, it is mention that nurses overburdened by supportive activities such as paper works, documentation, scheduling. However, such activities are parts of providers duty and they have role on quality and safe patient care provision. Do you think such tasks can be shifted to non-clinical staff ? Who can take over these supportive activities? Please add your opinion to the discussion or recommendation section.

Response: We provided our opinion in this regard in the ‘Recommendation’ section, paragraph 2, page 23, line 39-40: “If some of their support and additional activities can be shifted to other staff, nurses can devote their time better in nursing care.” We also addressed this in the same section, paragraph 4, page 24, lines 452-454: “Secondly, since the Nurses are found to be predominantly engaged in support activities at the expense of actual patient care, a separate staff category for administrative/ support activities is greatly warranted. This will free up the valuable yet scarce clinical time of the service providers.”

COMPETING INTERESTE:

WHO is the funder of this project and I see author/s from WHO, Bangladesh Office? Is it possible to declare that the authors have no competing interest?

Response: Thanks for identifying this issue. Colleagues from WHO have provided the following statement in response to the query: “This project was technically and financially supported by WHO Bangladesh. Authors have no conflict of interest to declare. The views expressed in this article are those of the authors. They do not necessarily represent the views of the organizations they are affiliated with.”

VERSION 2 – REVIEW

REVIEWER	Claire Su-Yeon Park Center for Econometric Optimization in the Nursing Workforce, Republic of Korea
REVIEW RETURNED	17-Dec-2019

GENERAL COMMENTS	To the authors, Thank you for your hard work. I did my best to give you constructive comments. I hope that my input is helpful for you to make your manuscript perfect. Below I outline an additional place the manuscript needs to be improved. Recommendations
--

Page 24, Line 454: "Thirdly, instead of the existing approach of deploying a fixed number of workforce at all health facilities, a flexible recruitment and HRH planning is needed, based on patient load and diseases burden. This can be supported by determination of absolute requirement of HRH in those health facilities." I suggested authors to revise the sentence in the previous review like this, "This can be support by determination of optimal requirements of HRM under given limited resources or constrains in those health facilities (please cite relevant references)." The reason why I asked you to revise it like that is that (1) the "absolute requirement" in the workforce planning cannot guarantee a patient-perceived reasonable quality of care, particularly, in a state of emergency and (2) the "absolute requirement" in the workforce planning is against your previous sentence (see a blue-colored phrase).

I confirmed that the authors followed my recommendation; however, I also found that the authors missed relevant citations. The sentence I recommended the authors to revise is based on up-to-date articles that have already been published in literature. The authors' work may risk research ethics if the citation problem would not be corrected.

I would like to advise the authors to perform additional literature reviews with a focus on the following articles:

Park, C. S. (2017). Optimizing staffing, quality and cost in home healthcare nursing: Theory synthesis. *Journal of Advanced Nursing*, 73(8), 1838-1847. doi: 10.1111/jan.13284

Park, C. S. (2018). Thinking outside the box. *Journal of Advanced Nursing*, 74(2), 237-238. doi:10.1111/jan.13312

Park, C. S., & Park, J. Y. (2019). Optimal safe staffing standard for right workforce planning. *Journal of Learning and Teaching in Digital Age*, 4(2), 42-44.

Saville, C. E., Griffiths, P., Ball, J. E., & Monks, T. (2019). How many nurses do we need? A review and discussion of operational research techniques applied to nurse staffing. *International Journal of Nursing Studies*, 97, 7-13. doi:10.1016/j.ijnurstu.2019.04.015

Lastly, "patient load and diseases burden" refers to "patient acuity" in literature. I would like to suggest the authors to use the well-defined scholarly term throughout the paper in a consist manner or provide the information in the brackets in its first use. Such effort will help readers a rapid grasp and an easier understanding of the manuscript.

My Second Review Comments

Dear Editor,

Thank you so much for giving me one more opportunity to review the manuscript entitled "An assessment of staffing needs for physicians and nurses at Upazila Health Complexes in Bangladesh using WHO workload indicators of staffing need (WISN) method." I carefully double-checked the manuscript and confirmed that authors sufficiently addressed concerns and issues I raised in the previous review (see below). Generally, I am

satisfied with this revision; however, I would like to present one additional place that the manuscript can be more improved as below. I humbly hope that my input can be of help to both BMJ Open and authors. Again, thank you for allowing me to review such a well-developed manuscript.

To the authors,

Thank you for your hard work. I did my best to give you constructive comments. I hope that my input is helpful for you to make your manuscript perfect. Below I outline an additional place the manuscript needs to be improved.

Recommendations

Page 24, Line 454: "Thirdly, instead of the existing approach of deploying a fixed number of workforce at all health facilities, a flexible recruitment and HRH planning is needed, based on patient load and diseases burden. This can be supported by determination of absolute requirement of HRH in those health facilities." I agree with your statement; however, I would like to suggest you a little more revision on the sentence I underlined, like this "This can be support by determination of optimal requirements of HRM under given limited resources or constrains in those health facilities (please cite relevant references)." The reason why I ask you to revise it like that is that the "absolute requirement" in the workforce planning cannot guarantee a patient-perceived reasonable quality of care, particularly, in a state of emergency.

My First Review Comments

Dear Editor,

Thank you so much for giving me the opportunity to review the manuscript entitled "An assessment of staffing needs for physicians and nurses at Upazila Health Complexes in Bangladesh using WHO workload indicators of staffing need (WISN) method." This manuscript perfectly falls within my specialization and research domain. I very much enjoyed reviewing this manuscript. This exciting experience inspired me to contemplate the scholarly identity, vision and promise of my own program of research. Even though I participated in this peer review process as a reviewer, I have benefited from such a well-delineated advanced scientific statement. I am truly honored beyond all measure to be a part of the panel of reviewers for your highly-reputed journal, BMJ Open.

This manuscript is significant in terms of (1) conducting a context-sensitive, time-motion study to provide a better understanding of the service context of the staffs in the real-world situations, based on the real-time data collection and analyses; (2) applying a bottom-up approach to determine activity standards (i.e. the authors estimated actual workloads); (3) considering an even distribution as well as an absolute number of physicians and nurses; and (4) revealing the novel finding that "nurses' maximum time is spent on administrative and paperwork tasks" (pp. 21, line 361-362), which has been less explored/addressed in the previous literature of the nursing workforce mainstream research. Based on the rationales, my recommendation is that BMJ Open would offer authors a revision opportunity.

To the authors,

Thank you for your hard work. I did my best to give you constructive comments. I hope that my input is helpful for you to improve the manuscript. Below I outline additional places the manuscript needs to be improved.

Introduction: Excellent description

Page 2, Line 29: Why four UpHCs, not 24 health facilities of Bangladesh? Please provide a core reason why authors reported the findings from the providers at four UpHCs only.

Page 2, Line 42: Why “there should be a gradual policy shift”? It seems illogical. Please consider to delete the sentence. Otherwise, paraphrase it, please.

Page 2, Line 43: What is “HRH”? Provide the full term at it first use, please.

Page 2, Line 44: Please insert “thus” in the following sentence: i.e. “WISN should thus be incorporated as a planning tool for health managers.”

Page 6, Line 130: What is “NGO”? Provide the full term at it first use, please.

Page 6, Line 124-135: Please provide a specific number of experts per category; e.g., “SC was selected from ** out of *** senior government officials, ** out of *** WHO personnel, and relevant academia (be specific. Which academia? How many experts were recruited from the relevant academia? Why?).” Also, please provide a rationale on your decision (i.e. justification) in terms of a number of experts and the experts’ specialization.

Page 7, Line 153: Is there any special reason why you chose the highest performing UpHCs?

Page 8, Line 166: Please suggest a possible specific rationale—e.g., working period/years—rather than your assumption, “assuming better knowledge of the activity standards,” to ensure a scientific integrity of the paper.

Page 8, Line 184: Please insert “or vice versa” in the end of the sentence.

Page 9, Line 186-190: Please provide specific examples per each category, i.e., health service activities, support activities, and additional activities.

Page 12, Line 236-243: Who is the principle experts and co-experts, respectively? Please specify them. Too many categorized participants involved in your study makes readers confused.

Page 21, Line 381-382: “Some staffs were not present at their service locations for various reasons.” Please provide the specific reasons or examples for BMJ Open readers’ easier and better understanding of the context of your study.

Discussion:

The authors virtually determined the differences between current and required staffing levels on the basis of actual activity

	standards (current) and healthcare professionals-perceived activity standards (required) only. That is, this study did not encompass patients' position (i.e. patient acuity or patient dependency) and stakeholders' stance (ensuring efficiency [cost] while guaranteeing effectiveness [quality of care]) in the healthcare workforce decision-making. The current research trend in the healthcare workforce resides in presenting informed shared decision-making rationales where parties of interests can be all satisfied in their decision-making. Patient engagement, patient reported outcome measures, precision medicine/nursing, creating shared value, and/or value chain are becoming more and more important for those reasons. All but nursing workforce studies rarely utilized well-verified specific workload indicators in the real-world situations such as the Workload Indicator of Staffing Need (WISN) method; thus, this study is significant and worthwhile. However, BMJ Open readers could benefit more from such a well-developed discussion while BMJ Open editors may find the manuscript of enough import to merit publication in the journal. On this, I would like to suggest the authors additional literature reviews on the issues I mentioned above and integrate them into the authors' discussion.
--	---

REVIEWER	Serebe Gebrie Transitional care of Las Vegas United States
REVIEW RETURNED	24-Dec-2019

GENERAL COMMENTS	This paper is a high-quality study that follows a Mixed study and involves experts in the field. This study contextually toiled the WHO tool(WISN) into Bangladesh setting. Many developing countries have similar problems. They are critically short in human resources in health care or their available health professionals are maldistributed. The current research and its methodology can be used as a baseline for further studies in developing and low-income countries.
---

VERSION 2 – AUTHOR RESPONSE

Reviewer 1:

Page 24, Line 454: “Thirdly, instead of the existing approach of deploying a fixed number of workforce at all health facilities, a flexible recruitment and HRH planning is needed, based on patient load and diseases burden. This can be supported by determination of absolute requirement of HRH in those health facilities.” I suggested authors to revise the sentence in the previous review like this, “This can be support by determination of optimal requirements of HRM under given limited resources or constrains in those health facilities (please cite relevant references).” The reason why I asked you to revise it like that is that (1) the “absolute requirement” in the workforce planning cannot guarantee a patient-perceived reasonable quality of care, particularly, in a state of emergency and (2) the “absolute requirement” in the workforce

planning is against your previous sentence (see a blue-colored phrase).

I confirmed that the authors followed my recommendation; however, I also found that the authors missed relevant citations. The sentence I recommended the authors to revise is based on up-to-date articles that have already been published in literature. The authors' work may risk research ethics if the citation problem would not be corrected.

I would like to advise the authors to perform additional literature reviews with a focus on the following articles:

Park, C. S. (2017). Optimizing staffing, quality and cost in home healthcare nursing: Theory synthesis. *Journal of Advanced Nursing*, 73(8), 1838-1847. doi: 10.1111/jan.13284

Park, C. S. (2018). Thinking outside the box. *Journal of Advanced Nursing*, 74(2), 237-238. doi:10.1111/jan.13312

Park, C. S., & Park, J. Y. (2019). Optimal safe staffing standard for right workforce planning. *Journal of Learning and Teaching in Digital Age*, 4(2), 42-44.

Saville, C. E., Griffiths, P., Ball, J. E., & Monks, T. (2019). How many nurses do we need? A review and discussion of operational research techniques applied to nurse staffing. *International Journal of Nursing Studies*, 97, 7-13. doi:10.1016/j.ijnurstu.2019.04.015

Response: Thanks for suggesting the references. We have now added them to our manuscript.

Lastly, "patient load and diseases burden" refers to "patient acuity" in literature. I would like to suggest the authors to use the well-defined scholarly term throughout the paper in a consistent manner or provide the information in the brackets in its first use. Such effort will help readers a rapid grasp and an easier understanding of the manuscript.

Response: There is a lack of consistency in the literature regarding how 'acuity' is defined and measured. One concept analysis reports that, "patient acuity is a measure of the severity of illness of the patient and the intensity of nursing care that patient requires" [Reference: Brennan, C.W. and

Daly, B.J. (2009), Patient acuity: a concept analysis. *Journal of Advanced Nursing*, 65: 1114-1126. doi:10.1111/j.1365-2648.2008.04920.x]. We agree with the reviewer that, it is imperative to use such scholarly terms with caution and consistency. Therefore, we decided not to use such a term as 'patient acuity' in the manuscript so as to avoid any confusion or contradiction. Instead of using a composite term like 'patient acuity', we used simple terms like 'patient load', 'disease burden', etc., separately.